# Variation of Cancer Incidence between and within GRELL Countries

**DOI:** 10.3390/ijerph18179262

**Published:** 2021-09-02

**Authors:** Paolo Contiero, Giovanna Tagliabue, Gemma Gatta, Jaume Galceran, Jean-Luc Bulliard, Martina Bertoldi, Alessandra Scaburri, Emanuele Crocetti

**Affiliations:** 1Environmental Epidemiology Unit, Fondazione IRCCS Istituto Nazionale dei Tumori, 20133 Milan, Italy; paolo.contiero@istitutotumori.mi.it (P.C.); martina.bertoldi@istitutotumori.mi.it (M.B.); alessandra.scaburri@izsler.it (A.S.); 2Cancer Registry Unit, Fondazione IRCCS Istituto Nazionale dei Tumori, 20133 Milan, Italy; 3Evaluative Epidemiology Unit, Fondazione IRCCS Istituto Nazionale dei Tumori, 20133 Milan, Italy; gemma.gatta@istitutotumori.mi.it; 4Tarragona Cancer Registry, Cancer Epidemiology and Prevention Service, Sant Joan de Reus University Hospital, IISPV, 43204 Reus, Spain; jaume.galceran@salutsantjoan.cat; 5Vaud Tumour Registry, Centre for Primary Care and Public Health, University of Lausanne, 1010 Lausanne, Switzerland; jean-luc.bulliard@unisante.ch; 6Neuchâtel-Jura Tumour Registry, 2000 Neuchâtel, Switzerland; 7Romagna Cancer Registry, Romagna Cancer Institute (IRCCS Istituto Romagnolo Per Lo Studio dei Tumori (IRST) “Dino Amadori”), 47014 Meldola, Italy; emanuelecrocetti@yahoo.com

**Keywords:** cancer incidence, geographic analysis, cancer registry, regional analysis, environmental justice, social inequalities

## Abstract

Variation in cancer incidence between countries and groups of countries has been well studied. However cancer incidence is linked to risk factors that may vary within countries, and may subsist in localized geographic areas. In this study we investigated between- and within-country variation in the incidence of all cancers combined for countries belonging to the Group for Cancer Epidemiology and Registration in Latin Language Countries (GRELL). We hypothesized that investigation at the micro-level (circumscribed regions and local cancer registry areas) would reveal incidence variations not evident at the macro level and allow identification of cancer incidence hotspots for research, public health, and to fight social inequalities. Data for all cancers diagnosed in 2008–2012 were extracted from Cancer Incidence in Five Continents, Vol XI. Incidence variation within a country or region was quantified as r/R, defined as the difference between the highest and lowest incidence rates for cancer registries within a country/region (r), divided by the incidence rate for the entire country/region × 100. We found that the area with the highest male incidence had an ASRw 4.3 times higher than the area with the lowest incidence. The area with the highest female incidence had an ASRw 3.3 times higher than the area with the lowest incidence. Areas with the highest male ASRws were Azores (Portugal), Florianopolis (Brazil), Metropolitan France, north Spain, Belgium, and north-west and north-east Italy. Areas with the highest female ASRws were Florianopolis (Brazil), Belgium, north-west Italy, north-east Italy, central Italy, Switzerland and Metropolitan France. Our analysis has shown that cancer incidence varies markedly across GRELL countries but also within several countries: the presence of several areas with high cancer incidence suggests the presence of area-specific risk factors that deserve further investigation.

## 1. Introduction

Cancer is a major cause of morbidity and mortality worldwide. It is estimated that in 2020 there were 19.3 million new cancer cases, 10.0 million cancer deaths, and 50.6 million survivors with cancer diagnosed up to 5 years previously [1]. Comparison of cancer incidence between countries and between areas within countries can reveal peaks and troughs indicating varying patterns of risk.

In this study we analysed cancer incidence in countries of Latin America, the Caribbean, and Europe, belonging to the Group for Cancer Epidemiology and Registration in Latin Language Countries (GRELL) [2]. GRELL countries have historical, cultural, and linguistic ties, but are widely separated geographically and have differing health systems and public health policies, also the risk factors for cancer show a very heterogeneous geographical distribution. For example the fraction (%) of all-cancers among males attributable to excess body mass index went in the GRELL countries from 1.3 (Ecuador) to 4.5 (Argentina), among females from 4.7 (Peru) to 11.6 (Puerto Rico) [3]. The ASRws attributable to all infectious agents, for males, went from 13.3 (Argentina) to 27.6 (Chile), for females from 20.4 (Uruguay) to 41.7 (Peru) [4]. The fraction (%) of melanoma cases among males attributable to ultraviolet (UV) radiation exposure went from 34.4 (Chile) to 94.4 (Switzerland), among females went from 1.20 (Peru) to 89.4 (Switzerland) [5]. Population weighted ambient concentrations (dust and sea-salt free) of fine particulate matter (PM_2.5_) between 2001 and 2010 went from 4.9 μg/m^3^ (Tropical Latin America) to some locations in the Po Valley in Italy exceeding the 35 μg/m^3^ [6]. Globally a positive relationship between overall cancer incidence and the Human Development Index, a composite index of life expectancy, education, and gross national income was observed, the same pattern was found in the GRELL countries. Usually the studies that compare cancer incidence at an international level compute the ASRws at a country level without considering within country variations [7,8,9,10]. Our aim was to assess whether all-cancer incidence varied between these countries, and also between geographic regions within countries, on the hypothesis that the investigation of regions and individual cancer registry areas would reveal information not evident at the country level. Identification of peaks and troughs indicating varying patterns of risk should stimulate etiologic research, provide indications for health resource allocation, and guide interventions to reduce risks [7,8,9,10].

## 2. Materials and Methods

### 2.1. Incidence

We downloaded cancer data pertaining to 108 GRELL cancer registries in 15 countries of Latin America, the Caribbean, and Europe. The more updated cancer incidence data, available on the IARC website [11] and published in Volume XI of Cancer Incidence in Five Continents [12], comprised cancer cases diagnosed in 2008–2012, together with person-years of follow-up, for each cancer site, by age at diagnosis (5-year age classes from 0 years), cancer registry, and sex. We computed age-specific incidence rates for every registry, for 5-year age classes (from 0–4 years to 75+ years), as the quotient of the number of cases and the number of person-years in the respective categories of sex and age-class, expressed per 100,000 person-years. We defined person-years as the sum of the population counts in a specific geographical area surveyed by a registry in the years for which registry data were published in Volume XI of Cancer Incidence in Five Continents, categorized by sex and age class. We estimated cancer incidence per 100,000 using the direct method standardized to the World Standard Population (ASRw), computing the weighted average of the age-specific rates using as weights the World Standard Population.

From these data we estimated cancer incidence per 100,000 using the direct method standardized to the World Standard Population (ASRw). This method is the same as that used by IARC [13], except that we merged age classes from 75 years on, because several registries only made data available for the merged ≥75 year age class. We estimated ASRws for all cancers combined, excluding non-melanoma skin cancer (ICD10 codes C00-C97, except C44) for each country, and also—when incidence variation was high as measured by the method of Crocetti et al. [13]—for geographic regions and sub-regions within countries. We computed the ASRw for countries and regions by using the same method.

### 2.2. Within-Country Incidence Variation

The variation in incidence within a country, or region within a country, was quantified as r/R, defined, after Crocetti et al., [13] as the difference (r) between the highest and lowest incidence rates for cancer registries within a country/region, divided by the incidence rate for the entire country/region × 100 (R). Crocetti et al. considered that an r/R of ≥30 indicated high (within-country) variation and that an r/R of <15% indicated low variation. We adopted the single r/R threshold of 30%.

The flow chart (Figure 1) illustrates how we proceeded.

We first examined incidence variation (r/R) between the cancer registries within a country (Step 1). When the r/R was <30% we used the overall incidence to represent the whole country. If the r/R was high (≥30%), we divided that country into geographic regions. In Step 2 we analysed the variation of incidence, by cancer registry, within each region. If a regional r/R was <30%, we used the regional incidence as representative of the whole region. If a regional r/R was ≥30%, we identified (Step 3) the cancer registry (or registries) responsible for the high r/R. Therefore, we report incidence rates for countries. However, if within-country incidence variation was high we report incidence rates for regions. Furthermore, if within-region variation was high we indicate the cancer registries responsible for the high variation.

## 3. Results

### 3.1. Breakdown of Countries into Regions According to r/R

Table 1A,B, for males and females respectively, show incidence (ASRw) for the 15 GRELL countries analysed, in order of decreasing incidence. Appendix A shows the list of the registries and the sizes of the populations under study by registry and by country. Four countries—Belgium, Costa Rica, Puerto Rico and Uruguay sent in data from nationwide cancer registries, so within-country variations could not be assessed, and only incidence by country is reported. Furthermore, for Portugal and Peru, data were only available for the Azores and Lima cancer registries, respectively, so again, within-country incidence variation could not be assessed. Of the remaining nine countries, seven countries for males were divided into regions because of high within-country incidence variation: Ecuador (r/R = 76.49%), Brazil (55.11%), Colombia (39.60%), Italy (35.13%), France (34.68%), Spain (30.30%), and Argentina (30.07%) (Appendix A). Switzerland (11.82%) and Chile (6.41%) were not divided into regions for low within-country incidence variations. For females, five of the nine countries were divided into regions because of high within-country incidence variation: Brazil (83.15%), Ecuador (68.35%), France (44.85%), Argentina (32%) and Italy (33.04%).

Spain (24.58%), Colombia (11.93%), Chile (5.81%) and Switzerland (11.82%) were not divided into regions for low within-country incidence variation (Appendix A). For males and females, Argentina [1] was divided into north, central and south; Brazil [2] into north-east, central, and south; Ecuador [3] into north-central, central, south, and Pacific; France [4] into Metropolitan and Overseas; and Italy [5] into north-east, north-west, central, south, and islands. For males only, Spain [6] was divided into north and east and south. As a result of these divisions (i.e., application of Step 2, Figure 1), 28 geographic entities in males (Table 2A) and 27 geographic entities in females (Table 2A) were identified, all now with low within-entity incidence variation.

Finally (Step 3, Figure 1) we isolated the cancer registries of Florianopolis (Brazil) for males and females and Cali (Colombia) for males only, resulting in 30 geographic entities in males and 28 in females.

### 3.2. Country and Regional Incidence Rates

#### 3.2.1. Males

Cancer incidence in males was higher in all European GRELL countries than all Latin American and Caribbean countries. Italy, the European country with lowest male incidence, was 40 ASRw points higher than Uruguay, the Latin-Caribbean country with highest male incidence (Table 1A). Azores (Portugal) had the highest male cancer incidence (381.92). The registry of Florianopolis (Brazil) had the second highest incidence (380.33), while Brazil was in the middle of the country rankings (272.58, Table 1A), and central and north-east Brazil were in the middle of the entity rankings. Incidence for Spain was 339.97 overall (Table 1A), with north and east Spain (Table 2A) at 357.91, and south Spain nearly 50 ASRw points lower at 308.34. As a country, France had the second highest male incidence (369.35). After dividing France into Metropolitan and Overseas areas, incidence in Metropolitan France remained high (372.25) while incidence in Overseas France was over 60 ASRw points lower than Metropolitan France. Belgium (364.28, country only) had a very similar incidence to the registries of France as a whole. Switzerland (338.22, country only) had similar incidence to Italy (331.81) but higher incidence than Brazil (258.28), while Florianopolis (Brazil) and north-west and north-east Italy had considerably higher incidence levels. Italy had the sixth highest incidence (Table 1A), and considerable incidence variation, with north-west and north-east Italy over 50 ASRw points higher than the Islands of Italy. Argentina (215.22) was split into high incidence central Argentina (222.12) and low incidence north Argentina (189.68) and south Argentina (173.24). As a country Colombia was near the bottom of Table 1A (179.15), but the cancer registry of Colombia-Cali had much higher incidence (204.49) than the Rest of Colombia (149.97). Costa Rica (173.90, country only) had a very similar incidence to the registries of Colombia as a whole. Ecuador had a similar pattern: as a country it had the lowest incidence in the GRELL area (135.11), but incidence varied widely: 192.79 for central-north Ecuador, 168.50 for south Ecuador, 125.43 for central Ecuador, and 89.45 for Pacific Ecuador.

#### 3.2.2. Females

As with males, cancer incidence was generally higher for females in European GRELL countries than Latin American/Caribbean countries (Table 1B). However there was an exception: Uruguay had higher incidence than Spain and Azores (Portugal). For Brazil, incidence for females was middle-ranking (199.34) but varied markedly by geographic entity: 338.43 in Florianopolis (highest in GRELL area), 208.57 in north-east Brazil, 194.18 in central Brazil, and 179.99 in south Brazil. Belgium had the highest country incidence (289.21, Table 1B) but Florianopolis was ranked at the top of Table 2B with an incidence nearly 50 ASRw points higher than Belgium. Italy (Table 1B) had the second highest incidence in females (264.22) but again with considerable geographic variation: from 281.59, 268.03 and 267.75, respectively, in north-east, central and north-west Italy, to 240.80 and 237.76 in south Italy and the islands. Female cancer incidence in Metropolitan France (262.20) and Switzerland (261.41) fell within the incidence range of the Italian regions, while Overseas France had low incidence (181.03). Spain, with middle-ranking female cancer incidence, was not split into regions because of limited within-country incidence variation. By contrast, Argentina was characterized by wide within-country incidence variation: 201.16 for the whole country (middle ranking), 207.66 for central Argentina, 174.77 for north Argentina, and 149.16 for south Argentina. Incidence for Costa Rica, Colombia, Ecuador, Peru (Lima only), Puerto Rico and Chile was low (bottom of Table 1B). For Colombia and Chile within-country incidence variation was low. Within Ecuador (152.43) there was marked variation: central Ecuador (144.88) and Pacific Ecuador (102.32) had low incidence; central-north Ecuador had much higher incidence (199.11).

#### 3.2.3. Comparisons

Table 3 shows ratios of highest to lowest ASRws for countries, regions, and cancer registries. For countries the ratios were 2.8 for males and 1.9 for females. For regions the ratios were 4.3 for males and 3.3 for females. For cancer registries the respective ratios were 4.8 and 3.3—closely similar to the regional ratios. The ten cancer registries with the highest all-cancer incidence in males, in order of decreasing incidence were Lille-Métropole (France), Loire-Atlantique (France), Doubs (France), Biella (Italy), Territoire de Belfort (France), Basque Country (Spain), Azores (Portugal), Florianopolis (Brazil), Sondrio (Italy) and Calvados (France). (Appendix A).

The 10 registries with the highest all-cancer incidence in females, in order of decreasing incidence were Florianopolis (Brazil), Parma (Italy), Romagna (Italy), Modena (Italy), Ferrara (Italy), Lille-Métropole (France), Belgium, Biella (Italy), Mantua (Italy), and Piacenza (Italy) (Appendix A).

## 4. Discussion

The aim of this study was to assess, by analysing data from 108 GRELL cancer registries in 15 countries of Latin America, the Caribbean, and Europe, the degree of variation in all-cancer incidence between countries, and also between geographic regions within countries, on the hypothesis that investigation of regions and individual cancer registry areas would reveal information not evident at the country level.

Crocetti et al. [13] found that neither 95% confidence intervals nor standard errors for national (or supra-national) incidence data provided reliable indications of within (or between) country variation in incidence, since both were largely dependent on the number of cases considered. In the present study, therefore, we used the r/R indicator proposed by Crocetti et al. [13] to identify marked within-country incidence variation and separate out the regions and individual cancer registries responsible for that variation.

Among the nine countries whose within-country incidence could be assessed, we found that seven countries for males and five countries for females could be divided into regions because of high within-country incidence variation, resulting in the identification of several high incidence areas (i.e., Florianopolis, Brazil; north and east Spain; north-east Italy) that were invisible when only incidence by country was considered. These findings suggest that our classification method is useful as it implies we succeeded in identifying the main incidence variations by isolating regions and cancer registries whose incidences differed most markedly.

To the best of our knowledge few studies have assessed all-cancer incidence variation between and within countries. Crocetti et al. [13] assessed all cancer incidence variation between and within Nordic countries. The study of Li et al. [14] investigated geographic variations in kidney cancer incidence in European countries, finding marked between and within country variations. The authors concluded that while between-country variations could be influenced by variation in diagnostic and clinical practice, but within-country variations were most likely due to yet-to-be-pinpointed environmental factors that require further investigation. Some studies tried to assess the site-specific cancer burden and their associated risk factors [15,16] using available country-level data for the population exposure to risk factors. The same data on risk factor distribution at a within country level were not currently easily available which limits the possibility to interpret some of the differences we observed without performing high-definition studies. However, some hypothesis could be suggested by looking at specific areas, for example, the differences in ASRw between north-east and north-west regions of Italy with respect to those of the south and the islands could be hypothesized to be linked to differences in diet, more Mediterranean in the south and the islands with respect to that of north-east and north-west [17] and to be linked to a different exposure of the populations to PM_2.5_ [6].

The most noteworthy finding of the present study is that the area covered by the Florianopolis (Brazil) cancer registry had particularly high all-cancer incidence, both in males and females: incidence was much higher than in other Brazilian regions, and up with the highest in Europe. Florianopolis is a costal and island city in the state of Santa Catarina in the south of Brazil. It has little heavy industry, and its economy is based mainly on information technology, tourism and services [18]. It is a wealthy area and it is possible that its residents may have more sedentary lifestyles, and consume more unhealthy foods, tobacco and alcohol, than populations of less wealthy parts of Brazil. However the exceptionally high rate suggests the possibility of a local environmental cause. Similar factors may account for the very high all-cancer incidence among males in the Azores, a group of volcanic islands in the middle of the Atlantic.

We also found high incidence for males in the cancer registry of Cali (204.49) compared to the rest of Colombia (149.97), however it is possible that Cali, the most ancient cancer registry in Latin America, suffered from less incompleteness of cancer registration than other cancer registries of the rest of Colombia [19,20,21].

Central-north Ecuador (192.79) also had high incidence compared with other regions of Ecuador (89.45–168.50). The north-central part of Ecuador corresponds in fact to the cancer registry that covers the city of Quito, the capital of Ecuador, and it is possible that many patients who do not normally reside in the city go there for treatment and this produces an over-registration with the corresponding artificial increase in incidence. These hotspots were not evident from nationally aggregated data and suggest the need for specific investigations that may reveal modifiable risk factors in these increased-risk populations.

We found that the ratio of the highest to lowest incidence in the GRELL regions we identified was 4.3 for males and 3.3 for females. These high ratios suggest that a large proportion of cases from high incidence areas are preventable. It has been estimated that between a third and a half of all cancers are preventable because they are due to modifiable risk factors (diet, lifestyle, environmental and occupational exposure to carcinogens and infectious agents), while non-modifiable factors, such as genetics, may only account for 4–5% of cases [16,22,23,24,25].

Social inequalities are among the factors likely to contribute to the within-country variations in cancer incidence we found. These inequalities, rooted in differences in income, education and opportunity, and also racial and sex discrimination, impact the most disadvantaged individuals, communities, and countries and strongly correlate with variations in cancer incidence, survival, and mortality between and within countries [25,26]. Social inequalities are also linked to different exposures to environmental carcinogens resulting in what has been termed environmental injustice [27,28,29,30,31,32]. Analyses that assess within-country variations in cancer incidence may, therefore, be a starting point for revealing such injustice.

A possible limitation of this study is that the cancer registries analysed may differ in completeness and data quality [19], however the data we used had been accepted by the IARC for CI5 after numerous data quality checks had been performed [33].

This main limitations of the present study arise from that fact that for several GRELL countries, incidence is available only in nationally aggregated form, rendering the identification of hotspots impossible. It is also impossible to identify hotspots in countries represented by cancer registries that cover just a small fraction of the area and population of those countries: The most conspicuous example in this study is Portugal which is represented solely by the small offshore registry of Azores. Old data suggest that incidence rates on mainland Portugal are much lower than those in the Azores [34]. We also emphasize that many of the countries investigated do not have nation-wide cancer registration so the within-country variations in cancer incidence we have documented are likely to be underestimates.

## 5. Conclusions

To conclude, this study has identified within-country regions whose all cancer incidence rates are three to four-fold higher than those in other regions in the GRELL area. It is likely that such differences (particularly in Italy, Spain, Colombia and Ecuador) are beyond differences in registration completeness and reflect differences in levels of industrialisation and economic development, which in turn increase occupational and environmental exposure to carcinogens and adversely affect lifestyle factors that increase cancer risk. For other high incidence areas such as Florianopolis and Azores, other explanations are possible and deserve further investigation. Our findings suggest that national cancer incidence statistics should also be published by region or individual cancer registry and not agglomerated at the country level. We think that studies investigating incidence for specific cancer sites have to be encouraged in the GRELL area.

## Figures and Tables

**Figure 1 ijerph-18-09262-f001:**
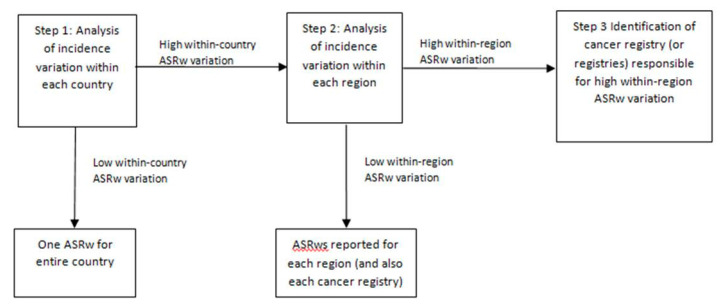
Flow-chart illustrating how geographic entities were defined according to variation in age-standardized incidence rates (ASRw).

**Table 1 ijerph-18-09262-t001:** Age-standardized incidence rates in males (A) and females (B) for all cancers combined (except non-melanoma skin cancer) in Group for Cancer Epidemiology and Registration in Latin Language Countries (GRELL) countries for the years 2008–2012. Rates are per 100,000 person-years and age-standardized to the world population (ASRw). Countries are listed in descending order of incidence.

1A		1B	
Country	Incidence Males	Country	Incidence Females
Portugal (Azores)	381.92	Belgium *	289.21
France	369.35	Italy	264.22
Belgium *	364.28	Switzerland	261.41
Spain	339.97	France	258.36
Switzerland	338.22	Uruguay *	216.77
Italy	331.81	Spain	210.56
Uruguay *	289.56	Portugal (Azores)	209.67
Brazil	272.58	Argentina	201.16
Puerto Rico *	258.28	Brazil	199.34
Argentina	215.22	Puerto Rico *	197.95
Chile	209.75	Peru (Lima)	187.51
Peru (Lima)	187.04	Chile	174.99
Colombia	179.15	Colombia	171.71
Costa Rica *	173.90	Costa Rica *	167.25
Ecuador	135.11	Ecuador	152.43

* Countries that sent in data from nationwide cancer registries, so within-country variations could not be assessed.

**Table 2 ijerph-18-09262-t002:** Age-standardized incidence rates in males (A) and females (B) for all cancers combined (except non-melanoma skin cancer) for geographic entities (countries, regions and cancer registries) characterized by low within-entity incidence variation (identified in Step 3, Figure 1). Incidence years are 2008–2012; rates are per 100,000 person-years, age-standardized to the world population (ASRw) and listed in descending order of incidence.

Region	Incidence Males	Region	Incidence Females
Azores, Portugal	381.92	Florianopolis, Brazil *	338.43
Florianopolis, Brazil *	380.33	Belgium **	289.21
Metropolitan France	372.25	North-east Italy	281.59
Belgium **	364.28	Central Italy	268.03
North-east Spain	357.91	North-west Italy	267.75
North-east Italy	346.53	Metropolitan France	262.20
North-west Italy	343.67	Switzerland	261.41
Switzerland	338.22	South Italy	240.80
Central Italy	324.55	Islands of Italy	237.76
South Italy	317.03	Uruguay **	216.77
South Spain	308.34	Spain	210.56
Overseas France	304.79	Azores, Portugal	209.67
Central Brazil	301.67	North-east Brazil	208.57
Islands of Italy	292.27	Central Argentina	207.66
Uruguay **	289.56	South Ecuador	206.51
North-east Brazil	281.88	Central-north Ecuador	199.11
Puerto Rico **	258.28	Puerto Rico **	197.95
South Brazil	235.58	Central Brazil	194.18
Central Argentina	222.12	Lima (Peru)	187.51
Chile	209.75	Overseas France	181.03
Cali,* Colombia	204.49	South Brazil	179.99
Central-north Ecuador	192.79	Chile	174.99
North Argentina	189.68	North Argentina	174.77
Lima (Peru)	187.04	Costa Rica **	167.25
Costa Rica **	173.90	Colombia	171.71
South Argentina	173.24	South Argentina	149.16
South Ecuador	168.50	Central Ecuador	144.88
Rest of Colombia	149.97	Pacific Ecuador	102.32
Central Ecuador	125.43		
Pacific Ecuador	89.45		

* Cancer registries responsible for high within-region incidence variation; ** Countries that sent in data from nationwide cancer registries, so within-country variations could not be assessed.

**Table 3 ijerph-18-09262-t003:** Ratios of highest-to-lowest incidence rate in GRELL countries, regions and cancer registries, by sex.

	Countries	Regions	Cancer Registries
Males	2.8	4.3	4.8
Females	1.9	3.3	3.3

## Data Availability

In this study we used only aggregated data available on web-sites accessible to the public.

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
