# Peer review of "Variation of Cancer Incidence between and within GRELL Countries"

_ijerph, 2021, doi:10.3390/ijerph18179262_

Round 1

Reviewer 1 Report

Manuscript ID: ijerph-1320124

Title: Variation of cancer incidence between and within GRELL countries

Authors: Paolo Contiero, et al

Submitted to section: Environmental Health

Please note that supplementary tables could not be accessed! I am willing to reevaluate my revision after receiving them.

The paper applies a method for evaluating variation in cancer incidence rates among areas. In particular it is applied using a hierarchical criterion (first countries, second regions, third registries) using the data of the large GRELL network of cancer registries.

The paper is clearly written. Rationale and results are well presented.

Some suggestions include:

  • Differences among countries also include external risk factors (carcinogens in the environment and in the occupational setting, prevalence of cancer related habits (eg smoking), dietary differences etc etc.) The text in lines 92 – 94 should be expanded. The discussion should also be expanded in this respect.
  • Please present in the introduction a comparison with other studies and methods of analysis of the variation of cancer rates (see line 99).
  • What is the rationale for the adopted criteria for subdivision of countries in regions? (see lines 158 and following)
  • The full list of country – regions – registries possible combinations should be clearly presented, either in the main paper or in the supplementary. This ‘full disaggregation table’ should also list the countries and regions and registries where no variation was detected, to provide the reader a full picture of the analyzed data. It is not clear to me if this information is entirely presented in table 2 (although it does not appear so, as registries with little variation are not included), if so please state it more clearly.
  • Main results are presented as to my understanding in table 2. However a few more information should be added for the reader for the interpretation of presented incidence figures. In particular: what is the population size of the areas? What is the 95% Confidence interval of presented incidence rates?
  • Please consider to sort results in table 2 according to a hierarchical order, corresponding to figure 1 (i.e. sort by country and region and registry).
  • Please expand the discussion to include also differences in exposure to carcinogens as an explanation of observed variation.

Are there any plans to repeat the analyses by cancer type? This point should be addressed in the discussion.

Author Response

REV 1

We thank rev1 for his precious suggestions that give us the possibility to improve our paper.

“Please note that supplementary tables could not be accessed! I am willing to reevaluate my revision after receiving them.

The paper applies a method for evaluating variation in cancer incidence rates among areas. In particular it is applied using a hierarchical criterion (first countries, second regions, third registries) using the data of the large GRELL network of cancer registries.

The paper is clearly written. Rationale and results are well presented.”

Some suggestions include:

“Differences among countries also include external risk factors (carcinogens in the environment and in the occupational setting, prevalence of cancer related habits (eg smoking), dietary differences etc etc.) The text in lines 92 – 94 should be expanded. The discussion should also be expanded in this respect.”

We expand the Introduction and Discussion section according to the suggestions of rev 1:

Added to the introduction:

“... also the risk factors for cancer show a very heterogeneous geographical distribution. For example the fraction (%) of all-cancers among males attributable to excess body mass index went in the GRELL countries from 1.3 (Ecuador) to 4.5 (Argentina), among females from 4.7 (Peru) to 11.6 (Puerto Rico) [3]. The ASRws attributable to all infectious agents, for males, went from 13.3 (Argentina) to 27.6 (Chile), for females from 20.4 (Uruguay) to 41.7 (Peru) [4]. The fraction (%) of melanoma cases among males attributable to ultraviolet (UV) radiation exposure went from 34.4 (Chile) to 94.4 (Switzerland), among females went from 1.20 (Peru) to 89.4 (Switzerland) [5].  Population weighted ambient concentrations (dust and sea-salt free) of fine particulate matter (PM2.5) between 2001 and 2010 went from 4.9 μg/m(Tropical Latin America) to some locations in the Po Valley in Italy exceeding the 35 μg/m3 [6]. Globally a positive relationship between overall cancer incidence and the Human Development Index, a composite index of life expectancy, education, and gross national income was observed, the same pattern was found in the GRELL countries. Usually the studies that compare cancer incidence at an international level computing the ASRws at a country level without considering within country variations [7-10].”

Added to the discussion:

“Many studies tried to assess the burden of cancer for specific sites and their associated risk factors [] using the available data at a country level for the population exposure to risk factors. The same data on risk factor distribution at a within country level are not currently easily available limiting  the possibility to interpret some of the differences we observed without making high-definition studies. However some hypothesis could be draw looking at specific areas, for example the  differences in ASRws between north-east and north-west of Italy respect to the south and isles areas of the same country could be hypothetized a relationship with differences in diet, more Mediterranean the one of south respect to the one of north []  and a differences with exposure to PM2.5 []”

“Please present in the introduction a comparison with other studies and methods of analysis of the variation of cancer rates (see line 99)”

We added the following sentence to the Introduction section:

“Usually the studies that compare cancer incidence at an international level compute the  ASRws at a country level without considering within country variations”.

“What is the rationale for the adopted criteria for subdivision of countries in regions? (see lines 158 and following)”

We used a geographical criteria following the subdivision in use in every country

“The full list of country – regions – registries possible combinations should be clearly presented, either in the main paper or in the supplementary. This ‘full disaggregation table’ should also list the countries and regions and registries where no variation was detected, to provide the reader a full picture of the analyzed data. It is not clear to me if this information is entirely presented in table 2 (although it does not appear so, as registries with little variation are not included), if so please state it more clearly.”

We apologise for the error, we did not upload into the IJERPH site the supplementary tables.

Now we uploaded them.

“Main results are presented as to my understanding in table 2. However a few more information should be added for the reader for the interpretation of presented incidence figures. In particular: what is the population size of the areas? What is the 95% Confidence interval of presented incidence rates?”

We added a supplementary table, S3, to list cancer registries and to show population sizes. We did not compute confidence intervals because our focus was on the variation indicator we proposed and we did not want to make the paper too complicated adding too much information

“Please expand the discussion to include also differences in exposure to carcinogens as an explanation of observed variation”

We added some sentences in the Discussion section.

“Are there any plans to repeat the analyses by cancer type? This point should be addressed in the discussion.”

We added the sentence in the Conclusion:

“We think that studies investigating incidence for specific cancer sites have to be encouraged in the GRELL area.”

Reviewer 2 Report

In my opinion the aim of the study is interesting but maybe it would be more valuable if  the authors  used the newest data or explained why they have chosen 2008-2012.  I think that the paragraph ,, Incindence" need to be decribed more in deatils since it is difficult to undrerstand for the readers.

Author Response

REV 2

We thank rev2 for his precious suggestions that give us the possibility to improve our paper.

“In my opinion the aim of the study is interesting but maybe it would be more valuable if  the authors  used the newest data or explained why they have chosen 2008-2012.  I think that the paragraph ,, Incindence" need to be decribed more in deatils since it is difficult to undrerstand for the readers”

We used the last data available on the IARC site for the “Cancer Incidence in Five Continents” publication and we added the specification that we used the more updated data in the Incidence paragraph.

We added to the Incidence paragraph the following details:

“We computed age-specific incidence rates for every registry, for 5-year age classes (from 0–4 years to 75+ years), as the quotient of the number of cases and the number of person-years in the respective categories of sex and age-class, expressed per 100 000  person-years. We defined person-years as the sum of the population counts in a specific geographical area surveyed by a registry in the years for which registry data were published in Volume XI of Cancer Incidence in Five Continents, categorized by sex and age. We estimated cancer incidence per 100 000 using the direct method standardized to the World Standard Population (ASRw), computing the weighted average of the age-specific rates using as weights the world standard population.”